# A Murine Model of Glioblastoma Initiating Cells and Human Brain Organoid Xenograft for Photodynamic Therapy Testing

**DOI:** 10.3390/ijms26188889

**Published:** 2025-09-12

**Authors:** Alejandra Mosteiro, Diouldé Diao, Carmen Bedia, Leire Pedrosa, Gabriela Ailén Caballero, Iban Aldecoa, Mar Mallo, Francesc Solé, Ana Sevilla, Abel Ferrés, Gloria Cabrera, Marta Muñoz-Tudurí, Marc Centellas, Estela Pineda, Àngels Sierra Jiménez, José Juan González Sánchez

**Affiliations:** 1Laboratory of Experimental Oncological Neurosurgery, Neurosurgery Service, Hospital Clinic de Barcelona—FCRB, 08036 Barcelona, Spain; dioulde.dg@gmail.com (D.D.);; 2Department of Neurosurgery, Hospital Clínic de Barcelona, Universitat de Barcelona, 08036 Barcelona, Spain; 3Institute of Environmental Assessment and Water Research (IDAEA-CSIC), 08034 Barcelona, Spain; carmen.bedia@idaea.csic.es; 4Department of Pathology, Biomedical Diagnostic Center, Hospital Clínic of Barcelona, University of Barcelona, 08036 Barcelona, Spain; gailen@clinic.cat (G.A.C.); ialdecoa@clinic.cat (I.A.); 5Neurological Tissue Bank of the Biobank, Institute of Biomedical Research August Pi i Sunyer (IDIBAPS), 08036 Barcelona, Spain; 6Microarrays Unit, Institut de Recerca Contra la Leucèmia Josep Carreras (IJC), ICO-Hospital Germans Trias i Pujol, Universitat Autònoma de Barcelona, 08916 Badalona, Spain; mmallo@carrerasresearch.org (M.M.); fsole@carrerasresearch.org (F.S.); 7Institute of Biomedicine, University of Barcelona (IBUB), 08036 Barcelona, Spain; 8Department of Cell Biology, Physiology and Immunology, Faculty of Biology, Institute of Neuroscience, University of Barcelona, 08028 Barcelona, Spain; 9Laboratorios Gebro Pharma S.A., 08022 Barcelona, Spainmarc.centellas@gebro.es (M.C.); 10Department of Oncology, Hospital Clinic de Barcelona—FCRB, 08036 Barcelona, Spain; epineda@clinic.cat; 11Department of Medicine and Life Sciences (MELIS), Faculty of Health and Live Sciences, Universitat Pompeu Fabra, 08036 Barcelona, Spain

**Keywords:** glioblastoma, photodynamic therapy, 5-ALA, organoid, model

## Abstract

Glioblastoma (GB) is one of the most aggressive brain tumors, characterized by high infiltrative capacity that enables tumor cells to invade healthy brain tissue and evade complete surgical resection. This invasiveness contributes to resistance against conventional therapies and a high recurrence rate. Strategies capable of eliminating residual tumor cells are urgently needed. Photodynamic therapy (PDT) using 5-aminolevulinic acid (5-ALA), an FDA- and EMA-approved compound, induces selective accumulation of the photosensitizer protoporphyrin IX (PpIX) in metabolically active tumor cells, enabling targeted cytotoxicity through light activation. A major limitation to its clinical application is the unclear variation in the cytotoxic effect of PDT according to individual tumoral differences. In this study, we propose and validate an in vivo model of patient-derived GB initiating cells (GICs) and brain organoids to test the effects of PDT. First, patient-derived GICs were molecularly characterized by flow cytometry and copy number variation profiling using OncoScan CNV Assays, then co-cultured with human brain organoids to generate a hybrid model recapitulating key aspects of the tumor microenvironment. 5-ALA photodynamic therapy (PDT) efficacy was assessed in vitro by GFP-based viability measurements, LDH release assays, and TUNEL staining. Then, a murine model was generated to study PDT in vivo, based on a heterotopic (renal subcapsular engraftment) xenograft of the GICs-human brain organoid co-culture. PDT was tested in the model; in each subject, one kidney tumoral engraftment was treated and the contralateral served as a control. Immunofluorescence analysis was used to study the cell composition of the brain organoid-tumoral engraftment after PDT, and the effects on non-GIC cells. The antitumoral effect was determined by the degree of cell death analysis with the TUNEL technique. The GICs-brain organoid co-culture resulted in tumoral growth and infiltration both in vitro and in vivo. The pattern of growth and infiltration varied according to the tumoral genetic profile. 5-ALA PDT resulted in a reduction in the number of GICs and an increase in apoptotic cells in all four lines tested in vitro. A correlation was found between the induced phototoxicity in vivo with the molecular typification of GICs cell lines in vitro. There were no changes in the number or distribution of neuronal cells after the application of PDT, while a reduction in active astrocytes was observed. 5-ALA PDT could be effective in eradicating GICs with a heterogeneous molecular profile. The hybrid human-murine model presented here could be useful in investigating adjuvant therapies in GB, under the concept of personalized medicine.

## 1. Introduction

Glioblastoma (GB) is the most common primary brain malignancy in adults, accounting for almost 80% of such tumors [1]. With the new 2021 World Health Organization Classification of Tumors of the Central Nervous System (CNS), molecular parameters have gained increasing relevance to define glioma subtypes [2,3], adding objectivity to diagnosis and improving patient stratification according to prognosis [4,5]. However, this implementation of tumor genetics has had little impact on the standard therapy for GB (Stupp protocol).

Unfortunately, recurrence is the norm, occurring on average seven months after treatment, invariably adopting a more aggressive form, and accounting for a one-year survival rate of approximately 60%. Notably, more than 80% of recurrences are located adjacent to the resection cavity. The fact that recurrences occur within the already treated area of resection denotes the ability of this tumor to resist conventional therapies. During the surgical resection, a certain degree of microscopic disease is left inadvertently; this is thought to correspond to the tumor front infiltrating white matter tracts and small vessels [6,7,8]. Several months after the surgery, this microscopic disease would be responsible for the tumoral recurrence due to senescence-induced changes after radiotherapy [9,10]. Thus, to prevent tumor recurrence, a plausible solution would be a tumor-selective locally delivered therapy, able to eliminate the microscopic disease left after surgical resection but avoiding damage to non-tumoral cells of the surrounding brain.

To serve this idea, photodynamic therapy (PDT) has been proposed. Indeed, after 5-aminolevulinic acid (5-ALA) administration, GB cells accumulate protoporphyrin IX (PpIX), a molecule that acts both as a fluorophore and a photosensitizer (PS). PpIX fluorescence can be visualized using a modified neurosurgical microscope and is already used for photodynamic diagnosis (PDD) [11,12].

5-ALA already represents a standard approach for intraoperative fluorescence-guided oncologic neurosurgery. Exploiting 5-ALA as both a fluorophore and a PS to visually identify the neoplastic tissue and, simultaneously, to selectively destroy it, could improve the success of GB treatment [13]. This synergistic mechanism of action, which integrates both therapeutic and diagnostic capabilities into one single product, has shown exciting prospects [14].

Although promising, data on the photo-toxic effects according to the diversity of genetic subtypes in GB is scarce [15]. Therefore, an experimental model to evaluate the tumor-sensitivity to the therapy is needed. Ideally, this model should mimic both the heterogeneity of the human GB and the interrelation with the brain-tumor microenvironment.

Our group has recently reported that glioblastoma initiating cells (GICs) co-cultured with brain organoids recapitulate many aspects of the disease and can be used in vitro to assess both the efficacy of local therapies and collateral damage to the surrounding brain cells [16]. The aim of the current study is to extrapolate this GIC-organoid co-culture into an in vivo murine model, in which to characterize the efficacy of PDT according to the genetic profile of GICs.

## 2. Results

### 2.1. Patient-Derived HCB-GICs Malignant Phenotype

Samples from patients operated on in the Hospital Clinic of Barcelona (HCB) were used to obtain HCB-GICs (Appendix A). All four selected patients were diagnosed with MGMT-methylated IDH wild-type grade 4 gliomas, following the WHO Classification of Tumors of the Central Nervous System criteria [2] by the Pathology Department. Mean age was 60.75 ± 11.44 years, with an equal distribution of 2 men and 2 women. Clinical details can be found in Appendix A. Progression-free survival (PFS) intervals varied widely between the four subjects, from early progression (95 days) to prolonged intervals (795 days).

Four different HCB-GICs (Figure 1A) were obtained from the patients’ samples: HCB-GIC1, HCB-GIC4, HCB-GIC19 and HCB-GIC20, with similar growth ratios but differing in their phenotype. After 15–20 days of isolation, HCB-GIC1, HCB-GIC19 and HCB-GIC20 formed tumorspheres, in contrast to HCB-GIC4, which grew in adhesion with a characteristic spicular cytoplasmic neural-like extension (Figure 1B and Appendix A). The stemness and invasive nature were further proven by evaluating the expression of specific markers (CD133 and CD44, respectively) with flow cytometry.

CD133 showed variable expression among HCB-GICs (Figure 1C). Whereas <20% of cells were CD133+ in HCB-GIC1 and HCB-GIC19, >70% of HCB-GIC4 and HCB-GIC20 cells were CD133+ (*p* < 0.0001). With 90% of cells across all four HCB-GICs being CD44+, no significant differences in CD44 expression were observed (*p* = 0.384) (Figure 1C). These findings confirmed that the aggressive phenotype is not always associated with the CD133 stem cell marker.

### 2.2. Patient-Derived HCB-GICs Genotype

We first explored the similarity between HCB-GICs obtained by primary culture of GB biopsies and their original tumor, to assess any deviation that could occur during the process of in vitro obtaining and expanding HCB-GICs (Appendix A). Shared imbalances were evaluated for their dimension (base pair length) and their representativeness (mosaicism percentage).

The comparison between DNA from the tumor tissue and the DNA from the HCH-GICs showed that HCB-GIC4, HCB-GIC19 and HCB-GIC20 shared almost 100% similarity. Interestingly, HCB-GIC1, the only one that differed from its primary tumor, had the highest accumulation of genetic alterations, suggesting genomic instability prone to induce in vitro genetic changes, resulting in the highest copy number alterations (CNA).

We looked for the three main canonical pathways described as being involved in glioma genesis and altered in high-grade gliomas: p53, Rb and PIK3KC A loss in p53 was seen in HCB-GIC1, and preserved in HCB-GIC4, HCB-GIC19 and HCB-GIC20 cells. CNA of HCB-GICs confirmed that only HCB-GIC1 had gains in both tumor suppressor genes, p53 and Rb, validating the clinicopathological diagnosis of the tumor samples. Since p53 pathway is more often modulated by several factors, including the upstream microtube-binding protein regulators as MDM1 and MDM2 [17], we checked if these genes were altered. However, our analysis indicated that both MDM1 and MDM2 remained normal.

HCB-GIC1 and HCB-GIC4 showed lost Rb; therefore, we examined genes such as CDK4 and CDK6, CDKN2A/B and CCND2 from Rb pathways. We found gains on CDK6 in all HCB-GICs with CDKN2 losses, while CDK4 and CCND2 gains were only observed in HCB-GIC1.

PIK3KC amplifications were found in HCB-GIC1, and PIK3KC was lost in HCB-GIC4. The PIK3KC pathway was further studied, including PIK3CA, PIK3R, PTEN, EGFR, PDGFR1 and NF1. HCB-GICs had exhibited PTEN losses, which were biallelic in HCB-GIC4 and HCB-GIC19. In addition, all HCB-GICs had EGFR amplifications, the highest 12 copies in HCB-GIC20. Moreover, PIK3CA and PIK3R were both amplified in HCB-GIC1. PDGFR1 was amplified in HCB-GIC1, lost in HCB-GIC4 and normal in both HCB-GIC19 and HCB-GIC20. NF1 was normal in all cases, except for HCB-GIC1, where the original tumor sample presented cnLOH loss, in mosaicism, whereas the GIC-derived showed only loss.

A genotypic heterogeneity among the four HCB-GICs was evident in the microarray analysis (Appendix A). HCB-GIC1 exhibited the highest number of alterations accumulated in the three main pathways, with 20 altered genes, followed by HCB-GIC4, with 10, and HCB-GIC19 and HCB-GIC20, both with 6 altered genes.

### 2.3. Organoid Cell Composition and HCB-GICs Differential Phenotype Determined Their Growth on Cerebral Organoid Co-Cultures

The expression of TUJ1 (neurons), GFAP (astrocytes), SOX2 (neural progenitor cells), and O4 (oligodendrocytes) in the brain organoids used in this study had been previously characterized and validated by immunofluorescence in our published work. That study demonstrated that these organoids reliably develop the main cell types found in neural tissue [16].

The pluripotent stem cell line BJ-Sev-iPSC possesses the intrinsic ability to self-organize into brain-like structures following sequential media changes. At 43 days, the multicellular aggregates contained polarized neural progenitors mimicking the developing cortex, as confirmed by the expression of key neural markers (Appendix A).

The pluripotent stem cell line BJ-Sev-iPSC owned the intrinsic ability to self-organize into brain-like structures after subsequent changes of media. At 43 days, the multicellular aggregate contained a group of polarized neural progenitors mimicking the developing cortex, with cells expressing neural, stem and glial cell markers. Immunofluorescence analysis on cryosection organoids confirmed the expression of Tubulin betta class 3 (TUBB3-TUJ1), a marker of neurons at an early stage of differentiation, and glial fibrillary acidic protein (GFAP), an astrocyte-specific marker (Appendix A). These validated brain organoids were subsequently used to establish HCB-GICs co-cultures.

HCB-GICs’ ability to infiltrate organoids was assessed after 30 days of co-culture to identify the depth of GFP-positive cells migration (Figure 2B). All HCB-GICs showed successful engraftment, with GFP+ cells growing within the organoid more than 100 µm from the surface, indicating the aggressiveness of the migration and invasion process (Figure 2A and Appendix A). The four HCB-GICs had different invasion features, with HCB-GIC1, HCB-GIC19 and HCB-GIC20 forming tumorspheres, and HCB-GIC4 disseminated within the organoid and triggering the characteristic spicular cytoplasmic neural-like extensions.

Moreover, we evaluated other malignant-associated phenotypic markers (Appendix A), the expression of vimentin and nestin proteins was analyzed [18]. Vimentin and nestin were generally expressed by the four HCB-GICs growing in brain organoids. The expression of SOX2 was analyzed in HCB-GIC co-cultures and detected in the four types, located mainly on tumoral foci. These results indicate that a stable malignant phenotype is maintained in the HCB-GICs co-cultured with brain organoids.

### 2.4. Photodynamic Therapy Effects In Vitro: HCB-GICs Infiltrating Brain Organoids

After 5-ALA PDT in vitro, a significant decrease in the GFP fluorescence between the irradiated and non-irradiated co-cultures of HCB-GIC1, HCB-GIC4 and HCB-GIC19 was observed after 24 h and 72 h, ranging between 0.4 and 0.6-fold decrease (*p* = 2.4 × 10^−3^, *p* = 5.3 × 10^−4^, and *p* = 2.1 × 10^−4^, respectively, at 24 h; *p* = 5.1 × 10^−3^, *p* = 1.5 × 10^−4^, and *p* = 1.5 × 10^−3^, respectively, at 72 h). In HCB-GIC20 co-cultures, a decreasing pattern in the fluorescence levels was also observed, although not statistically significant even 72 h after irradiation (Figure 3A,B and Appendix A).

A significant increase in cell death ratios was detected in the wells treated with 5ALA/PDT, as shown by LDH assay analysis (Figure 3C,D), compared to their untreated counterparts (Appendix A) after 72 h, in both HCB-GICs alone and those infiltrating organoids (HCB-GIC1, *p* < 1 × 10^−6^ and 6.2 × 10^−4^; HCB-GIC4, *p* < 1 × 10^−6^ and 3.5 × 10^−3^; HCB-GIC19, *p* = 1.5 × 10^−4^ and *p* = 5.6 × 10^−4^; HCB-GIC-20, *p* < 1 × 10^−6^ and *p* < 1 × 10^−6^). In wells containing only organoids, no increase in cell death was observed.

This pattern was confirmed by immunofluorescence, which revealed a visible reduction in fluorescent cells in HCB-GICs co-cultured organoids after PDT compared to non-treated controls (Figure 4A). Similarly, cell death analysis using TUNEL staining demonstrated a reduction in HCB-GICs accompanied by increased apoptotic signals after PDT in all four co-culture systems (Figure 4C).

Regarding tumor capacity to grow, infiltrate and resist apoptosis after PDT, we assessed the expression of vimentin, nestin and SOX2 by immunofluorescence. While the first two appeared unchanged after therapy, SOX2 expression showed an apparent decrease after irradiation (Appendix A).

### 2.5. Photodynamic Therapy Effects In Vivo: HCB-GICs-Brain-Organoid Xenografts

HCB-GIC20 and HCB-GIC1 infiltrating brain organoids (15 days of in vitro co-culture) were xenografted in the renal subcapsular space of both NSG mice kidneys (N = 3 in each group). Tumor and organoid neural tissue expansion were evaluated 40 days after engraftment.

PDT effectively decreased the HCB-GIC20 tumoral burden, measured as relative fluorescent units (RFU) on treated kidneys compared to contralateral untreated controls (*p* = 0.01) (Figure 5A,B). PDT proved to be effective even in cases of a big tumoral mass. Similar results were obtained with HCB-GIC1 tumors, where the average tumor RFU from control kidneys compared to that from treated kidneys was significantly different (*p* = 0.0483) (Appendix A).

Cell death was analyzed with the TUNEL technique (Figure 6A), which confirmed a statistically significant increase in apoptotic nuclei in treated cases, compared to untreated controls (*p* = 0.0371) (Figure 6B). The positive apoptotic nuclei were located within the necrotic areas.

To determine potential effects of PDT on the surrounding brain cells, the expression of neural (TUJ1) and astrocytes (GFAP) was analyzed in the engrafted kidneys by IF (Figure 6C,D). GFP indicative of tumor cells appeared surrounding necrotic areas, which were compatible with remaining brain organoid cells, where positive TUJ1 neurons are located. GFAP+ cells were observed in two different locations, in the middle of the necrotic areas and in the periphery of these areas, in close relation with tumor cells, suggesting reactive astrocytes. The quantity of these GFAP+ cells surrounding tumor cells diminished in the treated cases compared to untreated controls (*p* = 0.0071). GFAP+ cells were identified as astrocytes and not tumoral cells, as they did not express GFP.

## 3. Discussion

In this translational study, we evaluate the effects of 5-ALA-mediated PDT in an unprecedented experimental design of human glioblastoma, based on a co-culture of patient-derived HCB-GICs and human brain organoids, both in vitro and in an in vivo murine model. HCB-GICs were obtained from a series of patients operated on in our institution, demonstrating the potential of the model to study treatment effects under the concept of personalized medicine. According to our data, PDT seems effective both in vitro and in vivo. Moreover, the variations seen in the tumor cytotoxic effect according to the intrinsic genetic profile of each HCB-GIC subtype can be reproduced under in vivo conditions. The present design could be used to optimize preclinical studies of PDT and to explore adjuvant strategies to enhance 5-ALA phototherapeutic and photodiagnostic effects in gliomas.

### 3.1. A New Concept in GB Preclinical Investigation Models

Over the years, evidence generated by our laboratory and others has shown that patient-derived GICs are the most biologically and phenotypically relevant cells to study GB tumor initiation, maintenance and invasion [19]. The capacity for self-renewal and multi-lineage differentiation of GICs drives the increased resistance to ionizing radiation and cytotoxic drugs, limiting the effectiveness of current GB treatments [20]. The use of HCB-GICs for the development of tumor spheroids co-cultured with human brain organoids has provided a unique platform to characterize the behavior of GB, understand its relations with the tumor microenvironment, and test new therapies [21].

The introduction of organoids and tumorspheres as 3D models has enabled us to recapitulate more accurately the heterogeneity of the brain tissue and tumor properties [22]. These 3D constructions are obtained by culturing pluripotent stem cells in specific mediums that promote cell differentiation, and by subsequently co-culturing them with tumoral cells or spheres. Yet, they lack vascularization and are not able to recapitulate the systemic responses to tumor development and treatment regimes.

To overcome some of these limitations, we used a three-stage strategy to develop a valid model for evaluating the effects of PDT in GB. First, we obtained and characterized HCB-GICs from patients who had been operated on for GB. Second, we used these human HCB-GICs cell lines to infiltrate human brain organoids, containing neural, stem and glial cells. This 3D culture of HCB-GICs and brain organoids served as a platform to profile the malignant behavior of the different HCB-GIC subtypes, and to test their response to PDT with 5-ALA. Third, we transferred the HCB-GICs-organoid co-culture to the renal subcapsular space of NSG mice to test PDT in vivo.

We optimized the primary culture protocol, ensuring the maintenance of the initial HCB-GICs phenotype. The striking ability of HCB-GFP-GICs to proliferate and infiltrate the cerebral organoid simulated the infiltrative phenotype of GB. This cellular framework provided a potential mechanism for cell–to–cell signaling and intrinsic secretion of growth factors, guiding neuronal and glial development, altogether supplying a perfect microenvironment for HCB-GICs to invade and infiltrate organoids.

Only one of the cell lines showed significant genetic deviations from the original tumor, the HCB-GIC1. This underscores that the heterogeneity of GB might induce a sampling to in vitro disaggregation with different subpopulations, even with some advantageous cells able to develop subclones expanding the main genetic alteration and giving value to the model for research purposes.

### 3.2. Profiling Patient-Specific HCB-GIC Cell Lines

During the first stage of characterization of HCB-GICs derived from patients with GB, we found several differences in the genetic and molecular profiles. According to the consensus clustering that identifies four subtypes of GB, a high level of EGFR amplification is observed in 97% of the classical subtype and infrequently in other subtypes of GB [23]. EGFR amplification was found mainly in our HCB-GIC20 cell line. Deletions of NF1 with an altered Akt pathway, which characterizes the mesenchymal subtype, were found in our HCB-GIC1 line. Losses of PDGFR1, which are typical in the proneural subtype, were found in our HCB-GIC4 line.

CD133 is a membrane-bound glycoprotein seen on the surface of stem cells, used as a biomarker for GICs. CD133 is associated with the mesenchymal phenotype of GB cells, with poor prognosis, therapy resistance, and tumor recurrence [24]. We found CD133+ cells expressed mainly in the HCB-GIC4 and HCP-GIC20 lines. CD133 has been associated with the ability of tumor cells to self-renew and to grow in the form of aggregates (tumorspheres) in vitro [24]. In our sample, both CD133+ and CD133− cells were capable of forming tumorspheres, except for HCB-GIC4, which, despite being CD133+, did not demonstrate tumorsphere-forming ability and instead grew primarily in adherent conditions.

All HCB-GICs overexpressed CD44, which has been associated with the activation of genes mediating cell invasion under severe hypoxia conditions, and with genes mediating cell proliferation under moderate hypoxia. Both genetically activated pathways are key components of GB recurrence [25]. Our HCB-GIC cells also expressed the “stemness” markers Nestin, Vimentin and SOX2, which are known contributors to the invasiveness and infiltrative behavior of GB, and which allowed the tumorspheres to invade brain organoids.

### 3.3. Evaluating 5-ALA PDT Effect on HCB-GICs-Organoid In Vitro Model

Our group had previously explored the effects of 5-ALA-mediated PDT in vitro in a platform of GICs infiltrating cerebral organoids. In this initial experience, we demonstrated that two different GB cells, GIC7 and PG88, had different capacities to invade the brain organoid, and showed a different response to PDT [16].

At this stage, we used this model to test the effects of PDT in a series of HCB-GICs derived from patients operated on for GB. Here, we demonstrate that patient-derived HCB-GICs can invade the brain organoids and, importantly, remain susceptible to PDT. After the HCB-GICs-organoids were exposed to a medium containing 5-ALA and subsequently brought to a light source of 485 nm and 1.2 J/cm^2^, the burden of tumor cells in the organoids significantly decreased, while the cell death markers significantly increased.

Our findings support the idea that 5-ALA-mediated PDT can be an effective therapy for treating GB cells within the tumor niche. The model thus created could also be explored to study the potential adverse effects of PDT on the normal surrounding brain cells within the organoid. However, a limitation in this scenario is the lack of vascularization of the brain organoid. Indeed, the ability of GB to invade and induce the formation of neo vessels is a primary characteristic of this tumor [8]. This limitation may be overcome with an in vivo engraftment.

### 3.4. Evaluating 5-ALA PDT Effect on HCB-GICs-Organoid In Vivo Model

To better understand the limitations of the GICs-organoid in vitro model, we transferred the HCB-GICs co-cultured with human brain organoid to an experimental in vivo murine model. The spherical co-cultures of human GICs-organoids were implanted in the renal subcapsular space of immunosuppressed NSG mice [26]. This allowed us to observe the behavior and growth of the GICs-organoid in a vascularized environment. The design revealed a powerful in vivo system to recapitulate many aspects of GB development.

Compared to other murine models of GB, the unique content of both tumor cells and normal brain cells, both belonging to humans, makes this a unique platform. As both GB and neural/glial cells coexist within the same culture, it allows observing the interactions between transformed and non-transformed cells, making it useful to study essential tumor biology and giving valuable information for preclinical investigation of new compounds. For drug screening, this design enables an analysis of anti-tumor effects accompanied by collateral effects on healthy neural tissue, all in the same platform.

Optimizing PDT effects can be undertaken by increasing 5-ALA uptake, PpIX accumulation or exploring different light irradiation regimens that affect tissue oxygenation and may influence free radical production and oxidative stress. The importance of fractionation has already been emphasized by Curnow et al. and Vermandel et al. [27,28]. They found that the level of tissue oxygen at the treatment site was affected by the light regimens and that this indeed affected treatment effects. In our study, we opted for a rather simple irradiation scheme, as the main objective was to validate a model based on the combination of patient-derived GICs and human-derived glioneuronal organoids in which to test PDT. However, based on our encouraging findings, future experiments are warranted in which the effects of fractionated irradiation, controlled hyperoxygenation, or variable fractionated schemes based on PtiO2 values can be tested.

### 3.5. Strengths and Limitations of the GICs-Organoid Xenograft Model

In vivo models represent a step forward in modeling tumor complexity, as cells are implanted in a living organism where many types of dynamic interactions may occur, which condition therapy effectiveness. These may overcome some of the limitations seen in in vitro organoid models, in which the microenvironment interactions are restricted to cells that are already present in the tumor or that are artificially reconstructed and maintained by adding high amounts (even non-physiological) of cytokines, growth factors, serum, nutrients, etc.

An important feature in the HCB-GICs-organoid xenograft in mice is that tumoral and glioneural cells derive from humans and can be personalized with patient-derived tumoral and peripheral pluripotent cells. This advantage may be exploited to test the effects of specific therapies before administering them to the patient. Another remarkable asset of working with hybrid in vivo models based on organoid engraftment is the presence of vascularization and systemic immune responses, which may provide additional information regarding treatment response and mechanisms of resistance.

The limitations of this model are inherent to its xenograft nature and to its heterotopic location. The engraftment of human cells in mice requires the use of immunosuppressed strains, which do not allow studying the systemic immune responses to tumor development and to treatment. Conversely, it allows the use of patient-specific cells, which could have potential benefits in testing therapeutic responses in the era of personalized medicine. Furthermore, the idea of placing the tumor-organoid engraftment in the kidney allowed simplifying and homogenizing the experimental procedure and transforming the avascular tumor-organoid platform into a vascular one. However, this construction does not allow accounting for specific intracranial phenomena, such as the effects of the blood-brain barrier in the treatment response.

### 3.6. Immunological Implications of 5-ALA PDT Therapy

Recent studies have demonstrated that tumor-associated myeloid cells, particularly hyperactivated macrophages, exhibit elevated expression of heme oxygenase-1 and play a key role in ALA metabolism. These cells can accumulate and convert ALA into PpIX, suggesting a strong immunological and cellular component to PDT sensitivity and toxicity [29,30].

Although we did not specifically study myeloid cells in this study, we did find changes suggestive of immunological effects secondary to 5-ALA-mediated PDT. GFAP+ cells (astrocytes) were seen in the areas of infiltrative tumor. This astroglial proliferation/migration might be attributed to differences in the tumor microenvironment that promote astrocyte interactions with glioma cells and seem essential for glioma development and a crucial target in preventing glioma recurrences [16,31]. The role of astrocytes may change in the presence of a glial tumor [32]. Indeed, a distinct “reactive type” of astrocyte has been found in tumors, different from astrocytes purified from non-malignant specimens. This astrocytic change may be responsible for the impairment of immune responses, inducing a low reaction to novel therapies targeting the GB microenvironment [33]. The ability of astrocytes to survive local therapies is of particular interest, since they might mediate the antitumorigenic effect of these therapies and the potential collateral damage to the surrounding brain tissue [34].

In that sense, in our model, PDT decreased GFAP+ cells as well as HCB-GICs but did not seem to affect other glial and neural cells present in the organoid. These observations may indicate the relative specificity of the PDT treatment for both tumor and reactive astrocytes, while preserving the normal brain.

The modulation of the immunological response in the tumor microenvironment may explain the fact that PDT anti-tumoral effects seemed to be potentiated in vivo. In fact, the degree of tumor fluorescence in HCB-GIC20 cells was significantly higher in the in vivo xenograft than in the in vitro application of PDT. Following these preliminary observations, the presence of myeloid or other stromal components in the tumor microenvironment warrants further investigation in our model.

All in all, a major strength of this model, including human-derived glioneural organoids, is its potential ability to test collateral effects of damage to the surrounding brain when applying direct therapies such as 5ALA-mediated PDT in gliomas. Based on our preliminary findings, after the application of PDT, the number of TUT+ cells (corresponding to neurons) remained unchanged, while the GFAP+ cells (considered as reactive astrocytes) diminished in the area surrounding the tumor. These observations support the idea of PDT as tumor-specific and, up to some extent, immunomodulatory, yet this idea warrants future detailed exploration.

### 3.7. Limitations of the Study

Several limitations can be found in this study. First, the fact that the GIC cell lines derived from patient-specific samples limits the extrapolation of the therapeutic effects to the general population. Second, the therapeutic scheme for PDT in terms of irradiation time and dosage has not been standardized, and different results may be obtained with other regimes. Third, the effects attributed to the intracranial space, such as the presence of the blood–brain barrier or raised intracranial pressure, cannot be accounted for in this model. Fourth, the specific design in this study, where animal sacrifice was performed at the final stage after therapy, did not allow for a formal comparison of tumor volumes between the treatment and control kidney engraftments. However, given that the same number of GIC-organoids were implanted in each kidney and that they were exposed to the same conditions, a parallel tumor growth shall be assumed.

## 4. Materials and Methods

### 4.1. Obtention of Glioblastoma Cell Lines and Organoids

Glioblastoma-initiating cells (GICs) were obtained from surgical biopsies at the Hospital Clínic of Barcelona, as previously described by Frenster et al. (2018) [35]. All human-derived samples were collected with written informed consent under approval from the institutional Ethics Committee and in compliance with European and national biomedical research regulations. Detailed laboratory methods are provided in the Appendix A.

Cells transduction and Organoid generation. For longitudinal tracking, HCB-GICs were transduced with GFP-expressing lentiviral vectors [36]. Human iPSC-derived cerebral organoids were generated over 47 days in defined spheroid media with sequential growth and differentiation factors.

Co-culture assembly. Co-cultures were established by seeding GFP-labeled HCB-GICs onto mature organoids, as previously described by Pedrosa et al. (2023) [16]. Cultures were maintained for 20 days, with periodic monitoring of GIC infiltration using an Olympus BX41 microscope equipped with a GFP filter.

### 4.2. Molecular Profiling of HCB-GICs

DNA extraction and analysis. Genomic DNA was extracted from primary cultures by using the Blood and Cell Culture DNA Mini kit (Qiagen 13323, Hilden, Germany) and from tumor tissues by using the QIAamp DNA FFPE Tissue Kit (Qiagen, 56404, Hilden, Germany). Total DNA was quantified with the Invitrogen Qubit 4 fluorometer. Concentration and purity of DNA were determined by measuring the absorbance (A260/280) of the sample with the NanoDrop ND-1000 Spectrophotometer (Thermo Fisher Scientific, Waltham, MA, USA).

Genetic assessment. The OncoScan assays (Applied Biosystems™, Thermo Fisher Scientific) were used for genetic comparison between HCB-GICs and their respective tumours. Analysis was performed with Chromosome Analysis Suite Version 4.3 (Thermo Fisher Scientific).

### 4.3. Cellular Components of Human Brain Organoids and GICs Co-Cultures

The cellular composition and spatial distribution within the brain organoids were assessed by double immunofluorescence and by flow cytometry analysis of CD44 and CD133 expression in GFP^+^ cells (see Appendix A).

### 4.4. 5-ALA/PDT Testing In Vitro: HCB-GICs & Brain-Organoid Co-Cultures

The in vitro 5-ALA PDT experiments were conducted using twin 96-well plates: one plate for irradiation (1.2 J/cm^2^) and a paired non-irradiated control, as previously described by our group [16]. Each plate was split into two treatment groups—5-ALA (50 µg/mL) and vehicle control—each containing at least six replicates of (1) brain organoids, (2) HCB-GIC tumorspheres alone, and (3) co-cultures of organoids with HCB-GICs. To evaluate the four GIC lines, two sets of paired plates were used: one set for HCB-GIC1 and HCB-GIC4, and another for HCB-GIC19 and HCB-GIC20. After 24 h of 5-ALA exposure, one plate from each pair was irradiated for 15 min (total dose 1.2 J/cm^2^) while its twin remained non-irradiated.

### 4.5. Determination of PDT Efficacy In Vitro

In vitro efficacy of 5-ALA PDT was evaluated using three complementary assays. Fluorescence of HCB-GICs expressing GFP was measured (excitation 485 nm, emission 530 nm) from the bottom of each well (lid in place) at 12 randomized positions per well using an Infinite M Plex plate reader (Tecan, Trading AG, Switzerland). Readings were acquired immediately post-irradiation (t = 0) and at 24 h (t = 1) and 72 h (t = 2), and viability loss was quantified as the ratio of GFP signal at t = 1 or t = 2 relative to t = 0, with representative high-resolution images captured under GFP filter on an EVOS M7000 (Invitrogen, Thermo Scientific). Concurrently, 25 µL of supernatant was collected at each time point to measure lactate dehydrogenase activity (LDH Cytotoxicity Assay) (Promega, Madison, WI, USA). The amount of LDH released into the medium was used as an indicator of cell death. Differential LDH levels between time points were calculated to assess changes in cytotoxicity.

Apoptotic cell death was determined by TUNEL staining. Co-cultures were fixed and stained using the Click-iT Plus TUNEL kit (Invitrogen, C10618, Waltham, MA, USA) with Alexa Fluor 594, followed by DAPI counterstaining; fluorescent images were acquired on a Nikon Eclipse Ni-E microscope (Izasa Scientific, Barcelona, Spain) and analyzed with ImageJ 1.54g.

### 4.6. Animal Subjects

All experiments involving animals were approved by the Ethics Committee of our University (PNT 2090206 DOC/002). All methods were carried out in accordance with the institutional animal care guidelines and relevant regulations under the protocol CEEA 421/22, which were controlled periodically during the experimental process.

NOD/SCID/gamma(c)(null) mice (NSG), which are double homozygous for the severe combined immunodeficiency (SCID) mutation and interleukin-2Rgamma (IL-2Rgamma) allelic mutation (gamma(c)(null)), were used for the in vivo experiments.

### 4.7. 5-ALA/PDT Testing In Vivo: HCB-GICs Human-Brain-Organoid Xenograft

Details on the generation of the HCB-GICs brain-organoid xenograft in the renal subcapsular space of immunosuppressed mice can be found in Appendix A. After the engraftment, animals were kept alive for 40 days to allow for tumoral/organoid growth and infiltration prior to PDT testing. Specific details in the PDT procedure can be found in Appendix A.

Selected animals were administered 5-ALA (Gliolan, 5-aminolevulinic acid hydrochloride) orally at 100 mg/kg and protected from lighting. After 4 h, 5ALA/PDT was delivered to one of the kidneys (right side), at a wavelength of 635 nm and a power of 5 mW for 10 min. In each mouse, the contralateral kidney (left side) was left untreated as a control. The procedure was carried out under general anesthesia with isoflurane 4% induction and 2% maintenance, while oxygenation was kept at a FiO2 of 30% during the whole duration of the experiment. A custom-made integrated irradiation system was used for PDT (Universitat Politecnica de Catalunya, UPC, Barcelona, Spain). The instrument was specifically designed to guarantee that the light emitted was directed only to the area of interest. This was achieved by an adjustable diaphragm coupled with the light-emitting diode.

Before starting the irradiation period, the selected kidney was externalized by reopening the retroperitoneal incision used during the first intervention for tumor engraftment. The opening of the diaphragm was manually adjusted to cover the surface of the kidney, making sure the lightened area was restricted to the kidney. The contralateral kidney used as a control (and other surrounding tissues) was protected from the light source throughout the whole duration of the PDT delivery. For this, the control kidney was kept in the retroperitoneal space, and the area of light exposure was restricted to the area occupied by the contralateral externalized kidney.

After the treatment, animals were kept alive for seven days after sacrifice and kidney removal.

IVIS Imaging System (Caliper, Hopkinton, MA, USA) was used for ex vivo measurement of the relative fluorescent units (RFU) of GFP-positive cells as indicative of tumor burden. Emission of RFU on treated kidneys in comparison with the contralateral untreated controls was evaluated. Renal tissues were then fixed and preserved.

### 4.8. Statistical Analysis

Data are presented as mean ± SEM. Two-tailed Student’s t test or ANOVA were used as appropriate using GraphPad Prism (version 8.0.2, Insightful science, CA, USA). A Bonferroni post-hoc test was performed to compare the replicate means. Statistical significance was denoted as: * *p* < 0.05, ** *p* < 0.01, and *** *p* < 0.001.

Additional information on the methodology can be found in Appendix B, Appendix C, Appendix D and Appendix E.

## 5. Conclusions

We describe an in vitro and in vivo model of human glioblastoma based on a co-culture allowing the infiltration of glioblastoma-initiating cells into brain organoids, which can be studied ex vivo or transferred to immunosuppressed mice as a xenograft. Both strategies were constructed upon patient-derived tumoral cells and demonstrated to capture the genetic heterogeneity of glioblastoma, providing a realistic and personalized model for therapy testing. We successfully applied the model to test the effects of 5-ALA-mediated PDT both in vitro and in vivo and describe the differences in the treatment effects according to the genetic and phenotypic properties of each cell line. These results may be determinant to understand the role of the microenvironment in GB biology and treatment response.

## Figures and Tables

**Figure 1 ijms-26-08889-f001:**
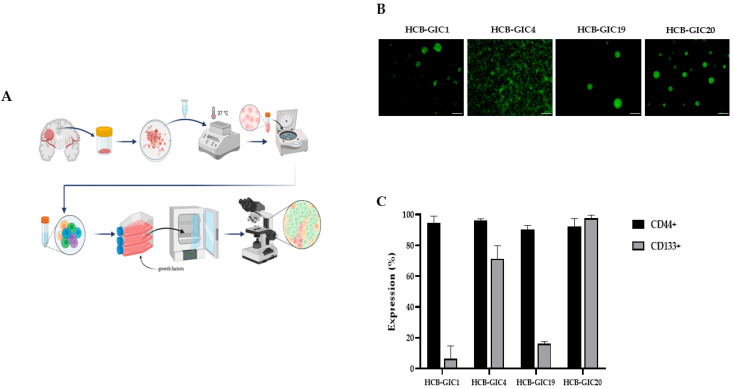
Establishment of Primary Human Glioblastoma tumorosphere cultures from biopsies. (**A**) Isolation protocol of HCB-GICs from human tissue samples. Schematic protocol to set up primary glioblastoma initiating cell (GICs) cultures. It includes steps from surgical human tissue samples until follow-up maintenance of GICs, as indicated in the scheme. (**B**) Four HCB-GICs were obtained: HCB-GIC1, HCB-GIC4, HCB-GIC19 and HCB-GIC20. Cultures of HCB-GICs constitutively expressing GFP are shown on maintenance conditions with complete NSC medium. (**C**) Flow cytometry analysis of CD133 and CD44 expression. HCB-GICs cells transfected with green fluorescent protein (GFP) were stained with anti-CD133 phycoerythrin (PE) or anti-CD44 labeled with AF594 as a secondary and subjected to flow cytometer analysis, collecting 10,000 events. Samples were analyzed on a Fortessa 5L flow cytometer. The figure is representative of three independent experiments. ANOVA test was performed to compare GFP+CD44+ (light gray bars) and GFP+CD133+ (dark gray bars) expression between the four cell lines. The results show that there are significant differences in CD133 expression between the cell lines, but not in CD44 expression.

**Figure 2 ijms-26-08889-f002:**
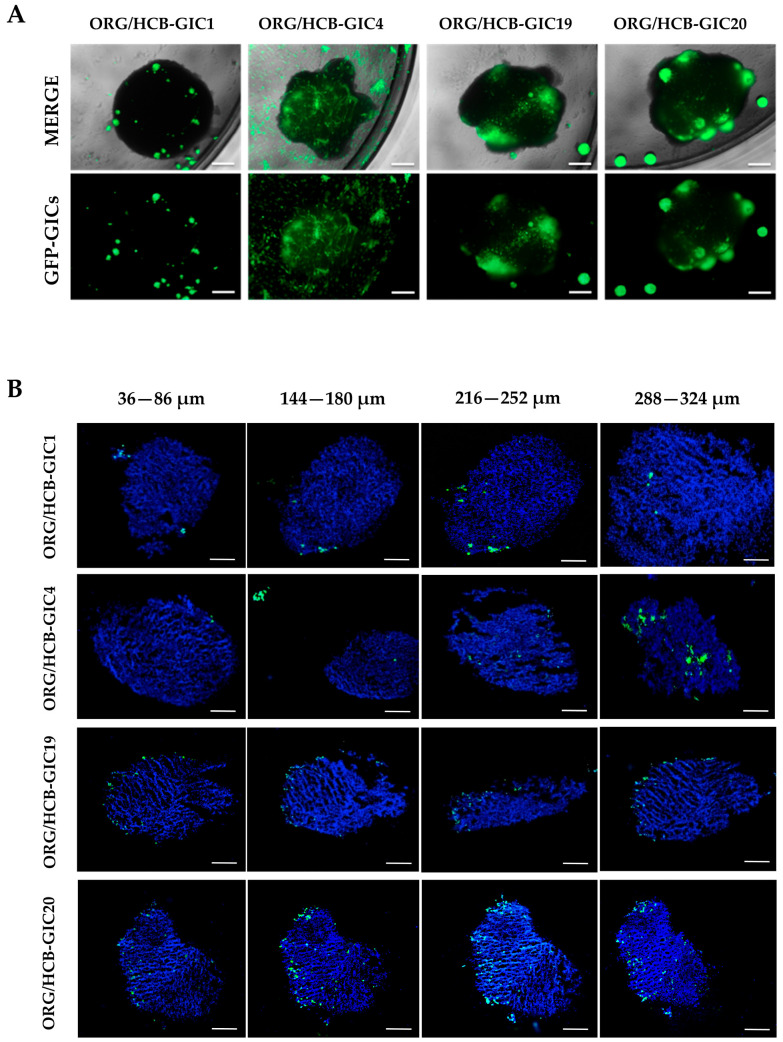
Start and progression of HCB-GICs and brain organoids co-culture. (**A**) Co-cultures were initiated on a 47-day-old brain organoid using 2 × 10^3^ cells per cell line. The experimental outline at 14 days post-initiation shows GFP-GIC infiltration (green fluorescence) patterns within the brain organoids (dark mass). HCB-GIC1, HCB-GIC19 and HCB-GIC20 have similar invasive disposition, but HCB-GIC4 show different behavior with characteristic spicular cytoplasmic neural-like extensions. Scale bars are: 100 µm. (**B**) HCB-GICs infiltrate brain organoids after 30 days of culture (N = 1 sample per condition). Fixed tissues were sliced at 12 μm. Nikon Eclipse Ni micrographs visualizing invasion of consecutive slides of frozen HCB-GIC organoid co-cultures show invasive GFP+ cells infiltrating the minibrain at indicated depths in the four models studied. Nuclei are counterstained with DAPI (blue). Scale bars are 100 μm.

**Figure 3 ijms-26-08889-f003:**
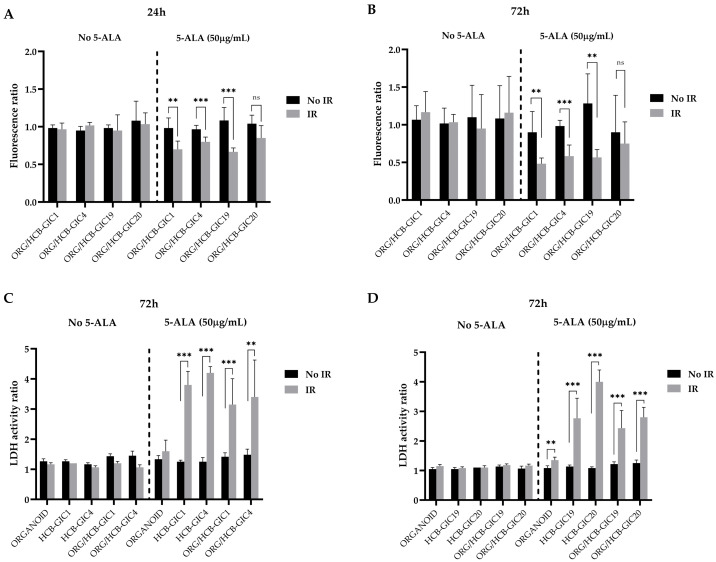
Photodynamic therapy in HCB-GICs infiltrating brain organoids. Two paired 96-well plates were used—one non-irradiated and one irradiated—with each plate split into two halves: medium alone and 5-ALA treatment (50 µg/mL). Each condition was performed in six replicates per column. GFP fluorescence ratios of HCB-GICs infiltrating organoids were calculated by dividing the fluorescence at 24 h (**A**) and 72 h (**B**) by the fluorescence measured immediately post-irradiation. Panels C and D show LDH assay results on supernatants collected 72 h after irradiation for HCB-GIC1 and HCB-GIC4 (**C**) and HCB-GIC19 and HCB-GIC20 (**D**). Statistical significance was determined by Holm–Sidak multiple comparisons test with ** *p* < 0.01, *** *p* < 0.001, ns (no statistical differences). See Discussion, Section 2.4 for detailed statistical analysis.

**Figure 4 ijms-26-08889-f004:**
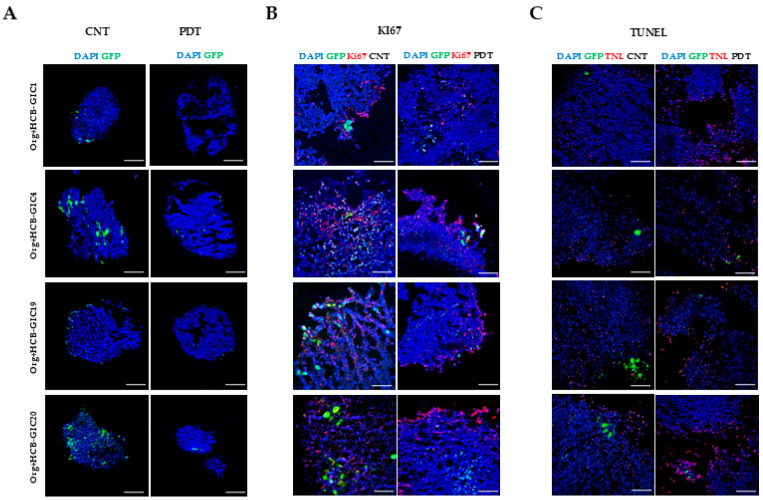
Immunofluorescence analysis of proliferation (Ki-67) and apoptosis (TUNEL) in PDT-treated co-cultures. Co-cultures were fixed at 30 days and cryosectioned at 12 µm (one section per condition) for qualitative immunofluorescence. HCB-GICs expressing GFP (green) invaded brain organoids and were divided into two groups: pretreated with 5-ALA (50 µg/mL) prior to PDT and untreated (control). Panels A–C show representative sections: (**A**) illustrates GFP^+^ GIC distribution in control and PDT-treated organoids (scale bars: 100 µm). (**B**) Panel B displays Ki-67 immunostaining (red) with DAPI counterstain (blue), comparing untreated controls (left) to PDT-treated samples (right) (scale bars: 20 µm). (**C**) Panel C shows TUNEL staining (red) under the same layout. PDT reduced Ki-67^+^ nuclei and increased TUNEL^+^ nuclei relative to controls. Images were acquired on a Nikon Eclipse Ni microscope and processed with ImageJ 1.54g. Color legend: Blue, DAPI (4′,6-diamidino-2-phenylindole); green, GFP (green fluorescent protein); red, ki67; black, PDT (photodynamic therapy).

**Figure 5 ijms-26-08889-f005:**
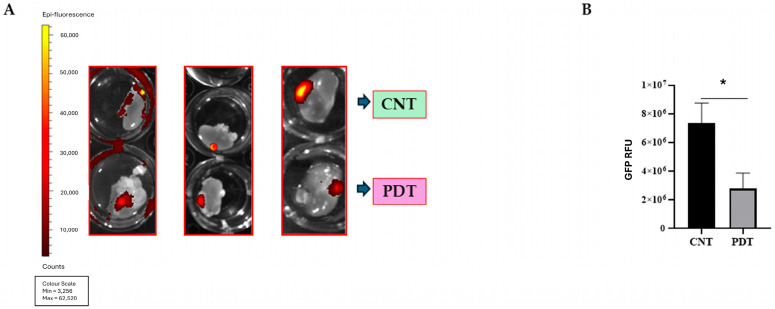
Ex vivo PDT efficacy in HCB-GIC20 organoid engraftment (N = 3 mice). Each mouse received two contralateral kidney implants: one control and one treated with PDT. At the endpoint, kidneys were harvested, and GFP fluorescence was measured ex vivo on an IVIS Imaging System (Caliper) to assess tumor burden. RFU values were corrected for local background and normalized to the number of implanted organoids. (**A**) Representative IVIS images of control (top row) and PDT-treated (bottom row) kidneys; the heatmap scale indicates RFU intensity. (**B**) Quantification of normalized GFP RFU in control versus PDT-treated kidneys. Data are mean ± SEM; unpaired two-tailed *t*-test performed in GraphPad Prism 8.0.2 on Fiji-processed images, * *p* = 0.0104.

**Figure 6 ijms-26-08889-f006:**
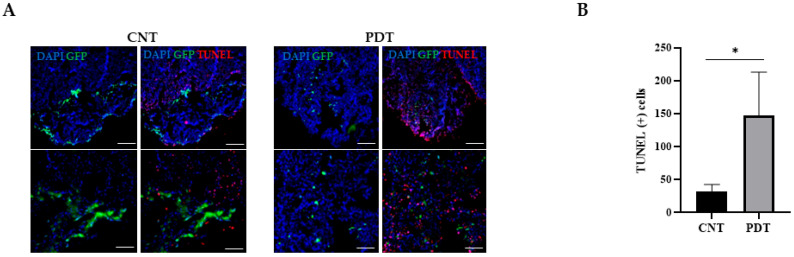
Immunofluorescence analysis of neuronal, glial, and apoptotic markers in HCB-GICs organoid grafts. Co-cultures were implanted into contralateral kidneys, with one kidney serving as an untreated control and the other subjected to PDT. (**A**) TUNEL staining (red) and GFP^+^ tumor cells (green) in HCB-GIC20 organoid grafts with DAPI counterstain (blue). Sections from untreated control kidneys show sparse apoptotic nuclei at the tumor periphery, while PDT-treated kidneys exhibit a marked increase in TUNEL^+^ nuclei coinciding with reduced GFP^+^ tumor density. Scale bars: 100 µm and 20 µm. (**B**) Quantification of TUNEL^+^ nuclei in five random 40× fields per sample (N = 3 mice) revealed a significant increase after PDT (unpaired two-tailed *t*-test performed in GraphPad Prism on Fiji-processed images, *p* = 0.0371). (**C**) TUJ1 (neurons, red) and GFAP (astrocytes, yellow) immunofluorescence with GFP^+^ tumor cells (green) and DAPI (blue). In controls, GFP^+^ cells infiltrate the organoid, TUJ1^+^ neurons outline its structure, and GFAP^+^ astrocytes are interspersed within and at the invasive front. PDT treatment markedly reduces GFP^+^ infiltration and GFAP^+^ astrocyte presence. Scale bars: 100 µm and 20 µm. (**D**) Quantification of GFAP^+^ area in five random 40× fields per sample (N = 3 mice) showed a significant decrease after PDT (unpaired two-tailed *t*-test performed in GraphPad Prism on Fiji-processed images, *p* = 0.0071; * *p* < 0.05, ** *p* < 0.01). Color legend: Blue, DAPI (4′,6-diamidino-2-phenylindole); green, GFP (green fluorescent protein); red, TUJ1 (beta-tubulin III), yellow, GFAP (Glial fibrillary acidic protein).

## Data Availability

Data may be shared with other investigators upon reasonable request and provided the approval of the Institutional Ethics Committee.

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
