# Peer review of "A Murine Model of Glioblastoma Initiating Cells and Human Brain Organoid Xenograft for Photodynamic Therapy Testing"

_ijms, 2025, doi:10.3390/ijms26188889_

Round 1

Reviewer 1 Report

Comments and Suggestions for Authors

The submitted manuscript describes the use of brain organoids that are cocultured with glioblastoma initiating cells for studies of treatment with 5-ALA mediated PDT.  The model that is described could be valuable in study of glioblastoma characteristics and therapeutic response.  Although interesting, the paper is lacking in scientific rigor. Major concerns are included among the numerous points listed below.  

Edits for English grammar are needed throughout the manuscript.  The examples provided below are representative of the edits needed, but not a comprehensive list.

Abstract:

It is not clear what is  meant by the sentence that ends with: “selective accumulation of photosensitizers after administration of 5-ALA for photodynamic therapy (PDT).”  ALA is not a photosensitizer itself, although it does lead to the endogenous production of protoporphrin IX.

Also not clear : “Still under development, variations in the cytotoxic effect of PDT according to individual tumoral differences are not defined.”

Overall, the  abstract needs to be reworded. It is not clear on the  experiment details and results.

Introduction:

Meaning unclear: “This is thought to derive from microtumor [6], [7] infiltration of vessels [8]and white matter tracts, which go inadvertently during surgery, and from senescence-induced changes after radiotherapy [9], [10]”

What is meant by “respecting the … brain?”

Results

“HCB” and “CNA” need to be defined.

Where are the data that accompany the results given in section 2.2 on genotype?

Where are the data to support the following claim in section 2.3:  “In all brain organoids, immunofluorescence analysis on cryosectioned organoids affirmed the expression of TUBB3 (neuronal marker), GFAP (astrocytes and radial glia), SOX2 (neural progenitor cells), and O4 (oligodendrocyte lineage cells), confirming proper neural differentiation.”

What does Figure S4 have to do with HCB-GIC invasion of organoids?  It doesn’t appear to be related to this.  

How many samples were evaluated per condition in the migration studies (Figure 2B).   Why are these data not quantified based on replicates?

Most immunofluorescence data appear to be observational. With the exception of Figure 6, these data are not quantified.

Figure S5 is referred to in the text as showing the results of PDT but the figure does not seem to show PDT.

Figures are not called out in order in the text.  They should be renumbered if needed in order to call them out in order.

Numbers of unique animals or samples needs to be indicated for each experiment, accompanied by summary statistics and appropriate testing.

The Materials and Methods contains many typographical errors.

It appears from Appendix D that PDT was delivered as a surface treatment to externalized kidney. What was the size of the treatment field? What irradiance was used and what dose was delivered?  

I cannot locate any legends for the figures or tables.

Comments on the Quality of English Language

English grammar must be improved.

Author Response

Reviewer #3: The submitted manuscript describes the use of brain organoids that are cocultured with glioblastoma initiating cells for studies of treatment with 5-ALA mediated PDT. The model that is described could be valuable in study of glioblastoma characteristics and therapeutic response. Although interesting, the paper is lacking in scientific rigor. Major concerns are included among the numerous points listed below.

Authors: We appreciate the thorough and critical revision of our work provided by the reviewer. We agree with his/her comments and believe that following the suggestions have helped us improve the quality of the work overall.

Edits for English grammar are needed throughout the manuscript. The examples provided below are representative of the edits needed, but not a comprehensive list.

Authors: A revision of the English language of the text has been made.

Abstract:

It is not clear what is meant by the sentence that ends with: “selective accumulation of photosensitizers after administration of 5-ALA for photodynamic therapy (PDT).” ALA is not a photosensitizer itself, although it does lead to the endogenous production of protoporphrin IX.

Also not clear : “Still under development, variations in the cytotoxic effect of PDT according to individual tumoral differences are not defined.”

Overall, the abstract needs to be reworded. It is not clear on the experiment details and results.

Authors: Following the reviewer’s indications, the abstract has been reformulated, as follows:

Abstract: Glioblastoma (GB) is one of the most aggressive brain tumors, characterized by high infiltrative capacity that enables tumor cells to invade healthy brain tissue and evade complete surgical resection. This invasiveness contributes to resistance against conventional therapies and a high recurrence rate. Strategies capable of eliminating residual tumor cells are urgently needed. Photodynamic therapy (PDT) using 5-aminolevulinic acid (5-ALA), an FDA- and EMA-approved compound, induces selective accumulation of the photosensitizer protoporphyrin IX (PpIX) in metabolically active tumor cells, enabling targeted cytotoxicity through light activation. A major limitation to its clinical application is the unclear variation in the cytotoxic effect of PDT according to individual tumoral differences. In this study, we propose and validate an in vivo model of patient-derived GB initiating cells (GICs) and brain organoid to test the effects of PDT.

First, patient derived GICs were molecularly typified by flow cytometry and copy number variation profiling using OncoScan CNV Assays, then co-cultured with human brain organoids to generate a hybrid model recapitulating key aspects of the tumor microenvironment. 5-ALA photodynamic therapy (PDT) efficacy was assessed in vitro by GFP-based viability measurements, LDH release assays, and TUNEL staining. Then, a murine model was generated to study PDT in vivo, based on a heterotopic (renal subcapsular engraftment) xenograft of the GICs-human brain organoid co-culture. PDT was tested in the model; in each subject, one kidney tumoral engraftment was treated and the contralateral served as a control. Immunofluorescence analysis was used to study the cell composition of the brain organoid-tumoral engraftment after PDT, and the effects on non-GICs cells. The antitumoral effect was determined by the degree of cell death analysis with TUNEL technique.

The GICs-brain organoid co-culture resulted in tumoral growth and infiltration both in vitro and in vivo. The pattern of growth and infiltration varied according to the tumoral genetic profile. 5-ALA PDT resulted in a reduction of the number of GICs and an increase in apoptotic cells in all the four lines tested in vitro. A correlation was found between the induced phototoxicity in vivo with the molecular typification of GICs cell lines in vitro. There were no changes in the number or distribution of neuronal cells after the application of PDT, while a reduction in activate astrocytes was observed.

5-ALA PDT could be effective in eradicating GICs with heterogeneous molecular profile. The hybrid human-murine model here presented could be useful in investigating adjuvant therapies in GB, under the concept of personalized medicine.

Introduction:

Meaning unclear: “This is thought to derive from microtumor [6], [7] infiltration of vessels [8]and white matter tracts, which go inadvertently during surgery, and from senescenceinduced changes after radiotherapy [9], [10]”

What is meant by “respecting the … brain?”

Authors: To improve the clarity of the introduction and following the reviewer’s indications, the abovementioned sentences have been changed as follows:

Introduction:

[…] The fact that recurrences occur within the already treated area of resection denotes the ability of this tumor to resist conventional therapies. During the surgical resection, a certain degree of microscopic disease is left inadvertently; this is thought to correspond to the tumor front infiltrating white matter tracts and small vessels [6] [7] [8]. Several months after the surgery, this microscopic disease would be responsible for the tumoral recurrence due to senescence-induced changes after radiotherapy [9] [10]. Thus, to prevent tumor recurrence a plausible solution would be a tumor-selective locally delivered therapy able to eliminate the microscopic disease left after surgical resection but avoiding damage to non-tumoral cells of the surrounding brain.

Results

“HCB” and “CNA” need to be defined.

Authors: The abbreviation HCB corresponds to the authors’ institution (Hospital Clínic Barcelona), at which the tumoral cells were obtained. The abbreviation CAN corresponds to Copy Number Alterations. Both have now been clarified in the results section.

Where are the data that accompany the results given in section 2.2 on genotype?

Authors: The genotype of each cell line is provided in Table 1 and in Supplementary Figure S2. We apologize for the Table not being included in the first version of the manuscript.

Where are the data to support the following claim in section 2.3: “In all brain organoids, immunofluorescence analysis on cryosectioned organoids affirmed the expression of TUBB3 (neuronal marker), GFAP (astrocytes and radial glia), SOX2 (neural progenitor cells), and O4 (oligodendrocyte lineage cells), confirming proper neural differentiation.”

Authors: An additional Supplementary figure (new Suppl. Fig. 4) has been added to show the cell composition of the differentiated brain organoids, containing all neuronal cells (TUJ1+) neuronal progenitor cells (SOX2), astrocytes (GFAP+), and oligodendrocytes (O4+).

Results: 2.3. Organoid cell composition and HCB-GICs differential phenotype determined their growth on cerebral organoid co-cultures 

The expression of TUJ1 (neurons), GFAP (astrocytes), SOX2 (neural progenitor cells), and O4 (oligodendrocytes) in the brain organoids used in this study had been previously characterized and validated by immunofluorescence in our published work. That study demonstrated that these organoids reliably develop the main cell types found in neural tissue [16].

The pluripotent stem cell line BJ-Sev-iPSC possesses the intrinsic ability to self-organize into brain-like structures following sequential media changes. At 43 days, the multicellular aggregates contained polarized neural progenitors mimicking the developing cortex, as confirmed by the expression of key neural markers (Supplementary Figure 4).

The pluripotent stem cell line BJ-Sev-iPSC owned the intrinsic ability to self-organize into brain-like structures after subsequent changes of media. At 43 days, the multicellular aggregate contained a group of polarized neural progenitors mimicking the developing cortex, with cells expressing neural, stem and glial cells markers. Immunofluorescence analysis on cryosection organoids confirmed the expression of Tubulin betta class 3 (TUBB3 - TUJ1), a marker of neurons at early stage of differentiation, and glial fibrillary acidic protein (GFAP), an astrocyte-specific marker (Supplementary Figure S4). These validated brain organoids were subsequently used to establish HCB-GICs co-cultures.

What does Figure S4 have to do with HCB-GIC invasion of organoids? It doesn’t appear to

be related to this.

Authors: The original Supplementary Figure S4 is showing the cell composition of the human brain organoid and the GICs co-culture under immunofluorescence analysis. GICs are marked with GFP (green fluorescent protein), while all cell nuclei are marked with DAPI (4′,6-diamidino-2-phenylindole). This is shown in the first column and allows to see how the GICs have infiltrated the brain organoid, as GFP+ cells are inside the organoid and dispersed in different areas.

In the second column an additional marker has been added, TUJ1, to demonstrate the presence of neuronal cells and their distribution in the organoid, and the relationship with tumoral cells. Neuronal cells are homogeneously distributed within the organoids, the tumoral cells seem to be infiltrating this neural tissue, but a clear pattern of cell-cell interaction is not seen with this IF analysis.

In the third column, a different marker is used, GFAP, to demonstrate the presence of astrocytes. GFAP is also a marker of glial cells, and some glioblastoma cells can be GFAP+. However, as tumoral cells have been marked with GFP, an assumption can be made that those cells GFAP+ and GFP- are astrocytes.

The fourth and fifth columns (the latter being a magnification of the former) contain all the above-mentioned markers. This combined IF analysis allows to see the distribution of the tumor and glioneuronal cells, which is interesting to understand de cell interactions and their behaviour in the tumor microenvironment. The pictures suggest that GFAP+ cells (reactive astrocytes) have migrated towards the areas of infiltrating tumour and that they tend to surround GFP+ (tumor cells) in these areas.

How many samples were evaluated per condition in the migration studies (Figure 2B). Why are these data not quantified based on replicates?

Authors: We evaluated one sample per condition, performing serial sections of 12 µm to assess invasiveness and determine whether infiltrating cells could be detected at comparable depths within the organoid structure. As the main goal of this assay was to qualitatively confirm the presence and distribution of invasive cells rather than to provide quantitative comparisons, we did not perform numerical analysis. We acknowledge that quantifying infiltrating GICs across conditions would be valuable for demonstrating potential differences in infiltration capacity, and we will consider this approach in future experiments.

Most immunofluorescence data appear to be observational. With the exception of Figure 6,

these data are not quantified.

Authors: The reviewer is right with this observation. The quantification of marked cells has only been done to serve the objectives of the present work, that is, to quantify the differences in the number of tumoral and glioneuronal cells in the GIC-organoids xenografts after PDT and to compare these figures with the untreated controls.

Therefore, the quantification of cells in the in vitro analysis of GIC-organoids co-cultures before engraftment, although interesting, was not contemplated in the study design.

Figure S5 is referred to in the text as showing the results of PDT but the figure does not

seem to show PDT.

Authors: The reviewer is right. Figure S5 is showing the infiltration of the organoids by the different GICs cell lines at baseline. The text has been modified to refer to Figure S6, in which the presence of GICs within the organoids is shown before and after the application of PDT in vitro and in the different cell lines.

To improve the understanding of Supplementary Figures, a description of each of them has been added to the supplementary material.

Figures are not called out in order in the text. They should be renumbered if needed in order

to call them out in order.

Authors: The number of the figures have been revised to guarantee an adequate order within the text.

Numbers of unique animals or samples needs to be indicated for each experiment, accompanied by summary statistics and appropriate testing.

Authors: We thank the reviewer for emphasizing the need to report sample sizes, summary statistics, and statistical tests. In response, we have revised all figure legends as follows:

  • Figure 1C now specifies that data are representative of three independent experiments. A one-way ANOVA with Tukey’s multiple comparisons was performed to compare GFP⁺CD44⁺ and GFP⁺CD133⁺ expression across the four cell lines (*p < 0.05; **p < 0.01; NS, no significant difference).
  • Figure 2B indicates that one biological sample per condition was analyzed (n=1), with tissues cryosectioned at 12 µm and imaged by Nikon Eclipse Ni; scale bars = 100 µm.
  • Figure 3 reports that two paired 96-well plates (irradiated vs. control), each split into medium and 5-ALA halves, were assayed in six technical replicates per column. GFP fluorescence ratios at 24 h and 72 h were calculated relative to  t=0, and LDH release was measured at 72 h. Data were analyzed by Holm–Sidak multiple comparisons test ( *p < 0.05). See Discussion, section 2.4 for detailed statistics.
  • Figure 4 clarifies that co-cultures were fixed at day 30, cryosectioned (one section per condition), and stained for Ki-67 and TUNEL. GFP⁺ organoid infiltrations in treated (5-ALA + PDT) and untreated controls are shown (scale bars: 100 µm and 20 µm).
  • Figure 5 now states n=3 mice, each bearing paired control and PDT-treated kidney implants. Ex vivo GFP RFU was background-corrected, normalized to organoid number, and compared by unpaired two-tailed t-test on Fiji-processed images in GraphPad Prism (mean ± SEM; *p = 0.0104).
  • Figure 6 specifies n=3 mice with contralateral grafts. TUNEL⁺ nuclei (five 40× fields/mouse) increased significantly after PDT (unpaired two-tailed t-test, p = 0.0371), and GFAP⁺ area (five 40× fields/mouse) decreased significantly (unpaired two-tailed t-test, p = 0.0071).

The Materials and Methods contains many typographical errors.

Authors: The English language of the Materials and Methods section has been revised. We apologize for the inconvenience.

It appears from Appendix D that PDT was delivered as a surface treatment to externalized kidney. What was the size of the treatment field? What irradiance was used and what dose was delivered?

Authors: We found this question interesting and needing further clarification. In fact, it was this issue that led us to design a specific device to administrate the light irradiation during the therapy, guaranteeing that all the surface of the kidney received a homogeneous dose of irradiation, and protecting the surrounding tissues (particularly the contralateral kidney serving as a control) from the direct light. To make this clearer, we have modified the methods section accordingly, as follows:

Methods, 4.7.:

Selected animals were administered 5-ALA (Gliolan, 5-aminolevulinic acid hy-drochloride) orally 100 mg/kg and protected from lighting. After 4h, 5ALA/PDT was delivered to one of the kidneys (right side), at a wavelength of 635 nm and a power of 5 mW for 10 minutes. In each mouse, the contralateral kidney (left side) was left un-treated as a control. The procedure was carried out under general anesthesia with isoflurane 4% induction and 2% maintenance, while oxygenation was kept at a FiO2 of 30% during the whole duration of the experiment. A custom-made integrated irradiation system was used for PDT (Universitat Politecnica de Catalunya, UPC, Barcelona, Spain). The instrument was specifically designed to guarantee that the light emitted was directed only to the area of interest. This was achieved by an adjustable diaphragm coupled with the light-emitting diode.

Before starting the irradiation period, the selected kidney was externalized by re-opening the retroperitoneal incision used during the first intervention for tumor engraftment. The opening of the diaphragm was manually adjusted to cover the surface of the kidney, making sure the lightened area was restricted to the kidney. The contra-lateral kidney used as a control (and other surrounding tissues) was protected from the light source throughout the whole duration of the PDT delivery. For this, the control kidney was kept in the retroperitoneal space, and the area of light exposure was restricted to the area occupied by the contralateral externalized kidney.

I cannot locate any legends for the figures or tables.

Authors: The legends for the figures have now been added at the end of the manuscript to facilitate the interpretation.

Reviewer 2 Report

Comments and Suggestions for Authors

This article does a nice job of demonstrating, from the bench to the murine study, the effects of 5-ALA-mediated photodynamic therapy on wild-type IDH glioblastoma cells.Several translational studies were fundamental in defining the parameters used for PDT (irradiation time, intervals, etc.). This served as a reference for the prospective series of patients treated with 5-ALA PDT, either as intraoperative adjuvance (treating the surgical bed immediately after resection) or as interstitial therapy for circumscribed unresectable tumors. These series showed benefits in disease-free survival and overall survival, delaying recurrence and aiding subsequent salvage treatments.

I have a few questions:
1. Even with the whole immunosuppressive and heterogeneous environment of glioblastoma, 5-ALA PDT tends to have a similar response in the different scenarios. This was very evident in the in vitro and in vivo results. The parameters used for the in vivo PDT were 635 nm/5 mW power for 10 minutes. Why was 10 minutes set? Could more cycles have been carried out, with intervals of 1-2 minutes for tissue reoxygenation? Also, was the O2 supply to the mice increased during PDT?

2. Considering that a treatment was carried out with a high fluence of light in a short irradiation time (clinical studies use up to 60' of treatment, divided into 12' cycles with 2' intervals), would it be feasible to test PDT in vitro with ultra-low fluence/longer time? 

3. In the in vivo study, was it possible to compare the light delivery between the kidney surface that received the 635nm light and the surface that had no contact with the light?

Author Response

We thank the reviewers and the editors for the time devoted to evaluating our manuscript. We have their impressions and suggestions with great interest and have tried to address all the points raised. Please, find below a response to each of them.

Reviewer #1: This article does a nice job of demonstrating, from the bench to the murine study, the effects of 5-ALA-mediated photodynamic therapy on wild-type IDH glioblastoma cells.Several translational studies were fundamental in defining the parameters used for PDT (irradiation time, intervals, etc.). This served as a reference for the prospective series of patients treated with 5-ALA PDT, either as intraoperative adjuvance (treating the surgical bed immediately after resection) or as interstitial therapy for circumscribed unresectable tumors. These series showed benefits in disease-free survival and overall survival, delaying recurrence and aiding subsequent salvage treatments.

I have a few questions:

  1. Even with the whole immunosuppressive and heterogeneous environment of glioblastoma, 5-ALA PDT tends to have a similar response in the different scenarios. This was very evident in the in vitro and in vivo results. The parameters used for the in vivo PDT were 635 nm/5 mW power for 10 minutes. Why was 10 minutes set? Could more cycles have been carried out, with intervals of 1-2 minutes for tissue reoxygenation? Also, was the O2 supply to the mice increased during PDT?

Authors: The questions raised by the reviewer about variables that may have an influence on the antitumoral effects of 5ALA-mediated PDT are indeed interesting. In fact, one of the main limitations to apply PDT is the fact that the optimal irradiation scheme is not yet defined. Some authors suggest that a fractionated irradiation may be more effective, as it allows for the reoxygenation of the tissue, similar to conventional radiotherapy. However, what is the optimal duration of the fractions? and the lapse between them? and the number of fractions? Vermandel et al. 2019 made an effort to solve these questions. In their experimental study, they concluded that a fixed fractionated scheme consisting of 5 fractions (16’ 40” ON followed by 2’ 30” OFF), 5mW 5J/fraction, 25J total was the most effective of all the studied schemes. However, these observations have not been validated by others.

The main objective of our study was to establish a robust model of glioblastoma, based on the combination of patient-derived GICs and human-derived glioneuronal organoids in which to test PDT. The model was tested with a rather simple scheme of irradiation, to prove its validity. The oxygenation was kept at a FiO2 of 30% during the whole duration of the experiment. However, future experiments are warranted, in which the effects of fractionated irradiation, controlled hyperoxygenation, or variable fractionated schemes based on PtiO2 values can be tested.

Following the reviewer’s question, we consider that these aspects merit inclusion on the discussion section of the paper, and we have thus designated a new paragraph, as follows:

Methods: “The oxygenation was kept at a FiO2 of 30% during the whole duration of the experiment.”

Discussion: “Optimizing PDT effects can be undertaken by increasing 5-ALA uptake, PpIX accumulation or exploring different light irradiation regimens that affect tissue oxygenation and may influence free radical production and oxidative stress. The importance of fractionation has already been emphasized by Curnow et al. et Vermandel et al. They found that the level of tissue oxygen at the treatment site was affected by the light regimens and that this indeed affected treatment effects. In our study, we opted for a rather simple irradiation scheme, as the main objective was to validate a model based on the combination of patient-derived GICs and human-derived glioneuronal organoids in which to test PDT. However, based on our encouraging findings, future experiments are warranted in which the effects of fractionated irradiation, controlled hyperoxygenation, or variable fractionated schemes based on PtiO2 values can be tested.”

REFS.

Vermandel M, Quidet M, Vignion-Dewalle AS, Leroy HA, Leroux B, Mordon S, Reyns N. Comparison of different treatment schemes in 5-ALA interstitial photodynamic therapy for high-grade glioma in a preclinical model: An MRI study. Photodiagnosis Photodyn Ther. 2019;25:166-176. doi: 10.1016/j.pdpdt.2018.12.003.

Curnow A, Haller JC, Bown SG. Oxygen monitoring during 5-aminolaevulinic acid induced photodynamic therapy in normal rat colon. Comparison of continuous and fractionated light regimes. J Photochem Photobiol B. 2000;58(2-3):149-55. doi: 10.1016/s1011-1344(00)00120-2.

  1. Considering that a treatment was carried out with a high fluence of light in a short irradiation time (clinical studies use up to 60' of treatment, divided into 12' cycles with 2' intervals), would it be feasible to test PDT in vitro with ultra-low fluence/longer time?

Authors: This question is in line with the previous one, remarking the importance of evaluating different irradiation schemes and fractionated irradiation, given that this may be an important modifier of treatment effects. Given the encouraging results obtained with our model, both at the in vitro and the in vivo phase, we expect to apply it to study different irradiations times and sequences, to try and solve the question of the optimal treatment regime in 5ALA-mediated PDT.

As indicated above, the discussion section of the manuscript has been modified to include this important parameter and to announce prospective lines of research in this line.

  1. In the in vivo study, was it possible to compare the light delivery between the kidney surface that received the 635nm light and the surface that had no contact with the light?

Authors: This point raised by the reviewer is also a very important one. The contralateral kidney used as a control was protected from the light source throughout the whole duration of the PDT delivery. The efficacy of PDT is highly dependent on the tissue exposure to the light source, and the effective depth of light penetration is usually limited to less than 10mm [1].

As mentioned in the Appendix D, a custom-made integrated irradiation system was used for PDT (Universitat Politecnica de Catalunya, UPC, Barcelona, Spain). The instrument was specifically designed to guarantee that the light emitted was directed only to the area of interest. This was achieved by an adjustable diaphragm coupled with the light-emitting diode. 

[1] Yassine AA, Lilge L, Betz V. Optimizing Interstitial Photodynamic Therapy Planning With Reinforcement Learning-Based Diffuser Placement. IEEE Trans Biomed Eng. 2021;68(5):1668-1679. doi: 10.1109/TBME.2021.3053197.

Following the reviewer’s pertinent question, we have now clarified this in the methods section, as follows.

Methods: “A custom-made integrated irradiation system was used for PDT (Universitat Politecnica de Catalunya, UPC, Barcelona, Spain). The instrument was specifically designed to guarantee that the light emitted was directed only to the area of interest. This was achieved by an adjustable diaphragm coupled with the light-emitting diode. The contralateral kidney used as a control was protected from the light source throughout the whole duration of the PDT delivery.”

Reviewer 3 Report

Comments and Suggestions for Authors

The manuscript presents a study of the efficacy of photodynamic therapy using 5-aminolevulinic acid on glioblastoma initiating cells in vitro and in vivo in murine models using GICs-human brain organoid. Results showed that PDT-induced phototoxicity in vivo correlated with the molecular characteristics of the GIC lines observed in vitro. The authors propose that the hybrid-human-murine model could be used in the future development of personalized therapies.

The authors provide a brief introductory section about glioblastoma and the importance of treating these tumors. They propose PDT using 5-ALA might be a solution to prevent the reminiscent tumor cells after surgical removal. The experiments are executed scientifically with all the details included. The authors have pointed out the limitations of this study in one of the sessions, this is appreciated. The major limitation noticed is the lack of appropriate figure and figure captions in the manuscript (detailed below):

Below are some of the specific suggestions/comments to the authors to address before manuscript publication:

  1. The figures are not included in the main manuscript which makes the review extremely difficult for the reader to correlate to the figures while reading the text
  2. Figures do not contain the figure captions. Please consider adding them for easy readability.
  3. Consider labeling Fig 1A for each step.
  4. Fig 2A, why is the fluorescence intensity observed even outside the main cell lines?
  5. Do authors predict any negative side effects of using 5-ALA for PDT in human cells?
  6. What are the next steps for this study that the authors would like to propose as the next step?

Author Response

We thank the reviewers and the editors for the time devoted to evaluating our manuscript. We have their impressions and suggestions with great interest and have tried to address all the points raised. Please, find below a response to each of them.

Reviewer #2: The manuscript presents a study of the efficacy of photodynamic therapy using 5-aminolevulinic acid on glioblastoma initiating cells in vitro and in vivo in murine models using GICs-human brain organoid. Results showed that PDT-induced phototoxicity in vivo correlated with the molecular characteristics of the GIC lines observed in vitro. The authors propose that the hybrid-human-murine model could be used in the future development of personalized therapies.

The authors provide a brief introductory section about glioblastoma and the importance of treating these tumors. They propose PDT using 5-ALA might be a solution to prevent the reminiscent tumor cells after surgical removal. The experiments are executed scientifically with all the details included. The authors have pointed out the limitations of this study in one of the sessions, this is appreciated. The major limitation noticed is the lack of appropriate figure and figure captions in the manuscript (detailed below):

Below are some of the specific suggestions/comments to the authors to address before manuscript publication:

  • The figures are not included in the main manuscript which makes the review extremely difficult for the reader to correlate to the figures while reading the text

Authors: We apologize for this inconvenience. Following the journal requirements during the submission process, figures were uploaded as a separated file. We have now tried to ensure that figures are available for revision.

  • Figures do not contain the figure captions. Please consider adding them for easy readability.

Authors: Again, we apologize for this inconvenience. We have now added all figure captions at the end of the manuscript so that they are readily available for revision.

  • Consider labeling Fig 1A for each step.

Authors: We agree with the reviewer that the stepwise scheme in Figure 1A would become clearer with a label set for each step. This has now been added to the figure.

  • Fig 2A, why is the fluorescence intensity observed even outside the main cell lines?

Authors: We appreciate this question, which merits clarification. Figure 2A represents the in vitro growth of a co-culture of GICs (marked with green fluorescent protein) and human brain organoids (not marked with fluorescent). Therefore, the green fluorescent seen here corresponds to tumoral cells (GICs). The presence of green outside the organoids corresponds to tumoral cells growing within the culture medium but not infiltrating the organoids.

Following the reviewer’s question, this has now been clarified in the figure caption.

Figure 2. Start and progression of HCB-GICs and brain organoids co-culture. A) Co-cultures were initiated on 47-day-old brain organoid using 2x103 cells per cell line. The experimental outline at 14 days post-initiation shows GFP-GIC infiltration (green fluorescence) patterns within the brain or-ganoids (dark mass. HCB-GIC1, HCB-GIC19 and HCB-GIC20 have similar invasive disposition, but HCB-GIC4 show different behaviour with characteristic spicular cytoplasmic neural-like extensions. Note that GICs are marked with GFP and human brain organoids are not marked; therefore, the green fluorescent seen here corresponds to tumoral cells (GICs). The presence of green outside the organoids corresponds to tumoral cells growing within the culture medium but not infiltrating the organoids.

  • Do authors predict any negative side effects of using 5-ALA for PDT in human cells?

Authors: This question about the collateral effects of PDT on the human glioneural cells surrounding the tumour is indeed very interesting. In fact, we believe one of the major strengths of our model is the ability to test this collateral damage specifically on human-derived cells.

According to our results, the cells positive to the marker TUT1 (corresponding to neurons) remained unchanged after 5-ALA PDT, indicating no phototoxic effect of the 5ALA PDT on neuronal cells. In contrast, GFAP-positive cells, representing astrocytes, decreased following treatment, particularly in the peritumoral region. This reduction could be attributed either to the death of GICs, which also express GFAP, or to an immunomodulatory effect of PDT that reduces the number of active/reactive astrocytes [1-3]. Thus the phototoxic action of 5-ALA PDT seems strictly limited to cells that have taken up the compound, have subsequently accumulated protoporphyrin IX, and fell within the illuminated area. It follows that any adverse effects on normal human brain cells are only likely if those cells had an alteration in the hem metabolism and are exposed to light. Our model minimizes this risk by employing a 4-hour dark incubation protocol before irradiation, favouring 5ALA uptake predominantly by highly metabolically active tumour cells, while limiting incorporation by normal cells such as astrocytes or neurons, which typically require longer exposure times. This selective uptake is further supported by our immunofluorescence analysis, which showed no signs of phototoxicity in TUJ1-positive neurons. The observed decrease in GFAP-positive cells is most likely due to the targeting and elimination of GFAP-expressing GICs rather than off-target damage to normal astrocytes

We consider that this idea raised by the reviewer merits inclusion in the discussion section, as so it has been addressed as follows.

Discussion: “All in all, a major strength of this model including human-derived glioneural or-ganoids is its potential ability to test collateral effects of damage to the surrounding brain when applying direct therapies such as 5ALA-mediated PDT in gliomas. Based on our preliminary findings, after the application of PDT, the number of TUT+ cells (corresponding to neurons) remained unchanged, while the GFAP+ cells (considered as reactive astrocytes) diminished in the area surrounding the tumor. These observations support the idea of PDT as tumor-specific and, up to some extent, immunomodulatory, yet this idea warrants future detailed exploration.”

  • What are the next steps for this study that the authors would like to propose as the next step?

Authors: In line with the reviewer’s suggestions and comments, we have added some future potential applications to the model here presented. As abovementioned, the idea of exploring the collateral damage to neuronal and glial cells surrounding the tumour has been considered.

Another point of interest for future research is the effect of fractionated irradiation, controlled hyperoxygenation, or variable fractionated schemes based on PtiO2 values. Different irradiation schemes and fractionated irradiation have been proposed by other authors as important modifiers of treatment effects. Given the encouraging results obtained with our model, both at the in vitro and the in vivo phase, we expect to apply it to study different irradiations times and sequences, to try and solve the question of the optimal treatment regime in 5ALA-mediated PDT.

These foreseeable lines of research have been added to the discussion section as follows:

Discussion: “All in all, a major strength of this model including human-derived glioneural or-ganoids is its potential ability to test collateral effects of damage to the surrounding brain when applying direct therapies such as 5ALA-mediated PDT in gliomas. Based on our preliminary findings, after the application of PDT, the number of TUT+ cells (corresponding to neurons) remained unchanged, while the GFAP+ cells (considered as reactive astrocytes) diminished in the area surrounding the tumor. These observations support the idea of PDT as tumor-specific and, up to some extent, immunomodulatory, yet this idea warrants future detailed exploration.”

Discussion: “Optimizing PDT effects can be undertaken by increasing 5-ALA uptake, PpIX accumulation or exploring different light irradiation regimens that affect tissue oxygenation and may influence free radical production and oxidative stress. The importance of fractionation has already been emphasized by Curnow et al. et Vermandel et al. They found that the level of tissue oxygen at the treatment site was affected by the light regimens and that this indeed affected treatment effects. In our study, we opted for a rather simple irradiation scheme, as the main objective was to validate a model based on the combination of patient-derived GICs and human-derived glioneuronal organoids in which to test PDT. However, based on our encouraging findings, future experiments are warranted in which the effects of fractionated irradiation, controlled hyperoxygenation, or variable fractionated schemes based on PtiO2 values can be tested.”

REFS.

Vermandel M, Quidet M, Vignion-Dewalle AS, Leroy HA, Leroux B, Mordon S, Reyns N. Comparison of different treatment schemes in 5-ALA interstitial photodynamic therapy for high-grade glioma in a preclinical model: An MRI study. Photodiagnosis Photodyn Ther. 2019;25:166-176. doi: 10.1016/j.pdpdt.2018.12.003.

Curnow A, Haller JC, Bown SG. Oxygen monitoring during 5-aminolaevulinic acid induced photodynamic therapy in normal rat colon. Comparison of continuous and fractionated light regimes. J Photochem Photobiol B. 2000;58(2-3):149-55. doi: 10.1016/s1011-1344(00)00120-2.

Round 2

Reviewer 3 Report

Comments and Suggestions for Authors

The authors have addressed all the comments and questions in detail. Very good job!